# District-Level Risk Factors for COVID-19 Incidence and Mortality in Nepal

**DOI:** 10.3390/ijerph19052659

**Published:** 2022-02-24

**Authors:** Dirga Kumar Lamichhane, Sabina Shrestha, Hwan-Cheol Kim

**Affiliations:** 1Department of Occupational and Environmental Medicine, School of Medicine, Inha University, Incheon 22212, Korea; carpediem@inha.ac.kr; 2Department of Community and Global Health, Graduate School of Medicine, The University of Tokyo, Tokyo 113-0033, Japan; sabinashrestha2001@yahoo.com

**Keywords:** COVID-19, district-level analysis, risk factors, Nepal

## Abstract

The recent global pandemic of the novel coronavirus disease 2019 (COVID-19) is affecting the entire population of Nepal, and the outcome of the epidemic varies from place to place. A district-level analysis was conducted to identify socio-demographic risk factors that drive the large variations in COVID-19 mortality and related health outcomes, as of 22 January 2021. Data on COVID-19 extracted from relevant reports and websites of the Ministry of Health and Population of Nepal, and the National Population and Housing Census and the Nepal Demographic and Health Survey were the main data sources for the district-level socio-demographic characteristics. We calculated the COVID-19 incidence, recovered cases, and deaths per 100,000 population, then estimated the associations with the risk factors using regression models. COVID-19 outcomes were positively associated with population density. A higher incidence of COVID-19 was associated with districts with a higher percentage of overcrowded households and without access to handwashing facilities. Adult literacy rate was negatively associated with the COVID-19 incidence. Increased mortality was significantly associated with a higher obesity prevalence in women and a higher smoking prevalence in men. Access to health care facilities reduced mortality. Population density was the most important driver behind the large variations in COVID-19 outcomes. This study identifies critical risk factors of COVID-19 outcomes, including population density, crowding, education, and hand hygiene, and these factors should be considered to address inequities in the burden of COVID-19 across districts.

## 1. Introduction

The spread of coronavirus disease 2019 (COVID-19), which is caused by severe acute respiratory syndrome coronavirus 2 (SARS-CoV-2), has evolved as a global pandemic [1]. As of 17 December 2021, more than 271 million confirmed cases were reported worldwide, and more than 5.3 million people have died due to COVID-19 [2]. This pandemic has posed further threats to people due to the emergence of the number of novel SARS-CoV-2 strains with unknown original hosts [3,4]. Several studies have been conducted to better understand the risk factors associated the spread and severity of COVID-19 infections. Research indicates that the risk of disease spread and deaths are influenced by several characteristics, such as socio-demographic factors [5], behavioral traits [6], and pre-existing health conditions [7]. However, the risk factors impacting the spread and severity of COVID-19 infections are inconsistent across studies, and they vary from place to place [8,9]. Therefore, studies focused on the local-level transmission of this disease are necessary for identifying the main drivers of disease spread that are suitable to contain the current pandemic in this specific region.

In Nepal, the first case of COVID-19 was confirmed on 23 January 2020 in a 32-year-old Nepalese man who had recently returned from Wuhan, China [10]. On 24 March 2020, the Government of Nepal implemented a strict lockdown, including business closures, restrictions on movement within the country, and flights in and out of the country [11]. The rate of spread of the disease was relatively low until mid-July 2020 [12], possibly because of the early nationwide lockdown. The government aggressively initiated a border screening policy to quarantine people traveling to Nepal from abroad, and provincial governments put in place targeted action on quarantine facilities and travel protocols. Nepal has faced multiple epidemic waves, with three distinct surge periods of COVID-19 cases: low (20 May to 25 June 2020), medium (22 July to 20 September 2020), and high (post-16 September 2020); these waves were due to an increase in susceptible population flow following the border opening (~20 May 2020), lockdown ending (~21 July 2020), and countrywide travel opening (~20 September 2020), respectively [13]. There was a rapid increase in the number of confirmed COVID-19 cases following the lifting of travel restrictions in many districts. As of 16 September 2020, a total of 58,327 cases were reported, and cases reached 268,948 on 22 January 2021 (end of the study) [14]. Despite a rapid spread of COVID-19, Nepal had high recovery rate, about 98%, as of 22 January 2021 [15], and a relatively low case fatality rate (CFR); the CFR was 0.6% up to 8 October 2020 [16].

The Nepalese government ended a country-wide lockdown on 21 July 2020 [13] and called for various preventive interventions on hand hygiene, health, and social distancing to be designed and implemented [17], possibly prioritizing areas at elevated risk. Washing hands with soap and running water is one of the best preventive measures to protect individuals and prevent the community from COVID-19 transmission [18]. In addition, population density and household crowding have emerged as important risk factors for COVID-19 transmission [8,9,19]. A previous study reported geographic variation in the pandemic trends in Nepal and suggested regional strategies along with the national-level strategy to control the local spread of COVID-19 [13]. Therefore, it is critical to understand the risk factors at the district level that are associated with widespread infection, severity of illness, and mortality. Representative data on the risk factors for COVID-19 mortality are lacking in Nepal. However, a recent nationally representative household survey and census data can be leveraged to find the risk factors for both the spread and severity of COVID-19 infections.

This study aims to examine the district-level socio-economic and demographic risk factors associated with the spread and severity of COVID-19 in Nepal. Identification of such risk factors can assist health policy makers in resource allocation decisions, provide evidence regarding the effectiveness of population health measures, and assist in developing a targeted, evidence-based response strategy to reduce the risk of subsequent waves of infection at a local level.

## 2. Materials and Methods

### 2.1. Data Extraction

Publicly available information on COVID-19-related health outcomes, consisting of the total number of cases, recovered cases, and deaths in all districts of Nepal, were assessed from the official websites of the different ministries of Nepal and previous studies [14,15,20,21]. For this study, we considered one year after the first case of COVID-19 was detected in Nepal (23 January 2020) as the final data capture point (22 January 2021).

Data for socio-demographic and health-related characteristics for each district were captured through various sources. These included per capita income based on purchasing power parity (2011) [22], the total population projection for 2021 [23], the age and gender distribution of the population [24], the population density (people per km^2^) [24], the adult literacy rate (2011) [22], and sanitation coverage (2011) [22].

For health indicators, we utilized data from the Nepal Demographic and Health Survey 2016 (NDHS 2016), a cross-sectional survey of 12,862 women and 4063 men in 11,040 households with a response rate of 98% of women and 96% of men. Details of the 2016 NDHS have been previously published [25]. Briefly, the survey was conducted from 19 June 2016 to 31 January 2017, and the sampling frame was based on the National Population and Housing Census 2011 (NPHC 2011), which was conducted by Nepal Central Bureau of Statistics [24]. The NPHC 2011 and NDHS 2016 were based on the 75 districts of Nepal. The eligible study population for the NDHS 2016 included all men or women aged 15–49 years who were permanent residents of the selected household or visitors who stayed the night in the households the night before the survey. The NDHS sampling and sample size were guided by the need to produce indicators that were representative at the district level. The survey included various socio-demographic, health, and family planning indicators. The survey used a two-stage stratified sampling design in rural areas and a three-stage design in urban areas. In both rural and urban areas, wards formed the primary sampling units (PSUs). Households were selected from the sample PSUs in rural areas, whereas one enumeration area (EA) was selected from each PSU in urban areas, and then households were selected from the sample EAs. PSUs were selected with a probability proportional to size, and households were selected using systematic sampling. We included only 73 districts for this study, as the health indicator data for the remaining two districts were not available.

### 2.2. COVID-19 Incidence and Mortality

The outcome variables of interest were the COVID-19 confirmed cases, recovered cases, and deaths as of 22 January 2021. For all districts of Nepal, we computed the total counts of confirmed cases per 100,000 persons (i.e., incidence), recovered cases per 100,000 persons, and deaths per 100,000 persons using the district-level population for the year 2021, which was projected from the census of 2001 and 2011 [23]. The district-level CFR was calculated by dividing the number of coronavirus deaths in the district by the total number of district cases.

### 2.3. Social Risk Factors

District-level social factors included population density, household crowding, sex ratio, proportion of elderly people, adult literacy rate, and per capita income. Population density was defined as persons per square kilometer in each district. Population density is considered as a proxy for the increased likelihood of crowded living environments and may increase the risk of COVID-19 transmission at a regional level [19]. According to the World Health Organization (WHO), household crowding is defined as the presence of more than three people per habitable room [26]. Household crowding has emerged as an important risk factor since spending a long period of time in close vicinity of an infected person significantly increases the risk of COVID-19 transmission [27]. In line with the WHO definition, we defined household crowding as the percentage of the households of a district who lived in homes with more than three people per room for sleeping. In our study, the percentage of household crowding was created using two variables from the household questionnaire of the NDHS 2016: the number of usual household members and visitors and the number of rooms used for sleeping. Furthermore, the sex composition of a population is indicated by the sex ratio, which was calculated as a ratio of total number of males to that of females multiplied by 100, indicating males per 100 females. In addition, we calculated the percentage of the total population of a district that was 60 years and above using the NPHC 2011. Older people are at higher risk of complications from COVID-19 [28]. In Nepal, 58.7% of deaths due to COVID-19 (data cutoff of 22 January 2021) were observed in elderly people (≥60 years) [15].

The level of literacy is a key social and economic indicator and has an important role in health communication. We obtained district-level adult literacy rates for 2011 from the Nepal Human Development Report by the United Nations Development Program (UNDP) [22]. The adult literacy rate was calculated as the ratio of literates, who can read and write, aged 15 years and above, by the corresponding age group of the population. In addition, the district-level per capita income in terms of purchasing parity per person (PPP) for 2011 was used to indicate the economic status of people and was obtained from the Nepal Human Development Report by the UNDP [22].

### 2.4. Factors Related to Hand Hygiene

Hand hygiene has become important in preventing the spread of COVID-19. In general, handwashing prevents germs from entering the body when people touch their eyes, nose, and mouth, as well as food and drinks [29]. The lack of access to handwashing facilities was defined as the percentage of the population of a district with no access to basic handwashing facilities at home, including soap and water [30]. We used the NDHS 2016 to estimate the percentage of the whole sample without basic handwashing facilities. The variable was constructed using three household questions related to handwashing facilities: (1) whether a handwashing facility was observed in the dwelling, (2) whether water was present at the handwashing facility, and (3) whether soap or detergent was present at the handwashing facility. We created a binary variable from these questions, and coded “1” if the handwashing facility was not observed or if there was either no water or no soap present and “0” otherwise. We estimated the absolute counts of people without access to handwashing facilities in each district. The percentage of people without access to handwashing facilities was calculated by dividing the absolute counts in each district by the number of the whole NDHS sample in each district multiplied by 100.

### 2.5. Health-Related Factors

The biomarker questionnaire of the NDHS 2016 collected measures of blood pressure (using an Omron Blood Pressure Monitor), height, and weight. We defined obesity as the percentage of eligible men or women (% of 15–49 years old) in a district with a body mass index, which was calculated as the ratio of weight in kilograms by the square of height in meters, equal or greater than 30. Hypertension was defined as the percentage of eligible men or women (% of 15 years and above) in a district that had systolic blood pressure > 140 mm Hg or diastolic blood pressure > 90 mm Hg. We excluded implausible values for blood pressures, such as systolic blood pressure above 250 or below 60 or diastolic blood pressure above 140 or below 40. We also obtained information on the access to health facilities at a district level using the NDHS 2016, which was defined as the percentage of households within 30 min walking distance of a government health facility. The access to a health facility reflects the availability of healthcare services in an emergency. In addition, we defined smoking as the percentage of men or women (% of 15–49 years old) who smoked cigarettes daily (manufactured or hand-rolled). The NDHS 2016 recorded daily cigarette smoking for all participants aged 15–49 years who were interviewed. In order to estimate the percentage of these risk factors (i.e., obesity, hypertension, access to a health facility, and smoking) at district level, we created a binary variable for each risk factor, indicating whether an individual experiences the risk factor. We calculated the percentage of a risk factor by dividing the number of people experiencing the risk factor in each district by the sample size for the risk factor in each district multiplied by 100.

### 2.6. Statistical Analysis

The unit of analysis was district in our analysis, and the baseline information on each district was reported as means, standard deviations, and proportions. A locally weighted scatterplot smoothing (LOWESS) curve was plotted to show the relationships between potential risk factors and COVID-19 outcomes per 100,000 people.

We applied a multiple linear regression to estimate the best fit regression equations and to assess the amount of variation that can be explained by the risk factors, assuming independent noise terms, all with an identical normal distribution. We built several main effects multivariable regression models to identify the factors significantly associated with the COVID-19 cases, recovered cases, and deaths per 100,000 people. In order to reduce overfitting caused by the limited sample size (n = 73 districts), the potential predictors for model development were first identified by a univariable screening process with a pre-set *p*-value of 0.25. This approach is recommended for removing weak predictors [31]. Then, we used a backward stepwise elimination approach, based on a likelihood ratio test, to select the final set of covariates for retention in the COVID-19 outcome models. All models were robust to heteroskedasticity (Breusch–Pagan test), and multicollinearity was not observed, as measured by the variance inflation factor (VIF < 3).

A Poisson regression model is typically used to evaluate count data. In our study, an initial assessment of the outcome variables indicated considerable overdispersion, meaning that the variance exceeded the mean. Therefore, a negative binomial regression model was used to estimate the rate ratio (RR) of the risk factors for COVID-19 outcomes. The outcome variables were the total number of COVID-19 cases, recovered cases, and deaths. The population size for each district was included as an offset to estimate the standardized RR: a value < 1 indicates a decreased likelihood and a value > 1 indicates an increased likelihood of the event under investigation. We assessed overdispersion in each model using a likelihood ratio test, which compares the negative binomial model to a Poisson model. A statistically significant *p*-value for chi-square with one degree of freedom indicates the presence of overdispersion.

Furthermore, we employed restricted cubic spline models to examine potential non-linear associations between the covariate that explain large variations in the linear regression model, COVID-19 case rates, and other outcomes. To provide enough flexibility to the model and to make the model less sensitive to the smallest fluctuations, we prespecified the use of three knots [32].

Additionally, to validate the model, we randomly split the data into training and validation sets using a 60/40 split and evaluated the model in both the training and validation sets. Adjusted R^2^ and McFadden’s pseudo R^2^ were used to assess the model performance in the linear regression and negative binomial regression, respectively. All the statistical analyses were conducted using Stata v.17.0 (Stata Corp., College Station, TX, USA).

## 3. Results

The descriptive statistics of the outcome and exposure variables for the 73 districts are summarized in Table 1. As of 22 January 2021, the average rates for case fatality and recovery were 1.2% and 97.2%, respectively. The proportion of the population aged 60 and above was 6.1%, and it ranged between 1.8% and 13.7% among the districts of Nepal. The population density (number of persons per square kilometer) varied from 4.7 in Dolpa district to 4415.8 in Kathmandu district. Household crowding, across districts, ranged between 3.3% and 37.9%, while the adult literacy rate ranged from 16.0% to 66.1%. A wide variation across districts was observed for the prevalence of individuals who did not have access to a handwashing facility (mean 60.9%; range 9.8% to 100%). Furthermore, the prevalences of obesity in women and smoking in men were 3.6% and 18.9%, respectively.

Appendix A presents confirmed, recovered, and deceased cases by district for the entire study period (23 January 2020 to 22 January 2021). As of 22 January 2021, 268,948 COVID-19-positive cases were reported, with 263,546 recovered, and 1986 deaths. The confirmed cases of COVID-19 were distributed throughout the country in all the administrative districts. Among the 73 districts included in the analysis, the total number of confirmed cases was highest in Kathmandu district (n = 103,523), followed by Lalitpur (n = 16,106) and Morang (n = 13,236) districts, and was lowest in Mugu (n = 37), Humla (n = 44), and Dolpa (n = 60) districts. The highest number of cases was reported in the age group 21−40 years (53.18%, n = 143,039) (Figure 1A); however, the number of deaths was higher in the age group 61−80 (Figure 1B).

In Nepal, the overall rate of cases, recovered cases, and deaths were 881.04, 863.35, and 6.51 per 100,000 people (Appendix A), indicating that 11.18 cases per 100,000 people had continuing illness when the data collection was stopped. The distribution of COVID-19 outcomes varied across districts in Nepal (Figure 2). The top 10% of districts by incidence accounted for 36% of the cases, whereas the lowest 10% of districts by incidence accounted for only 1.8% (Figure 2A). The highest COVID-19 incidence was 4499 per 100,000 (Appendix A). The top 10% of districts by mortality accounted for 35% of all COVID-19 deaths (Figure 2C). The highest rate of COVID-19 mortality was 32 per 100,000 (Appendix A). In addition, we conducted an analysis using province-level data to show interprovincial variation in the COVID-19 outcomes. Among the provinces, Bagmati province had the highest average rate of confirmed cases (1191 per 100,000), recovered cases (1164 per 100,000), and deaths (9 per 100,000) in Nepal, followed by Gandaki and Lumbini (Figure 2D–F).

The COVID-19 incidence and mortality per 100,000 are plotted as a function of district-level predictors using the bivariate smoother (“LOWESS”) for each predictor (Figure 3). The five predictors, including population density, obesity in women, sex ratio, no access to handwashing facilities, and per capita income, showed positive associations with the COVID-19 incidence and mortality per 100,000 people.

The findings of the multivariable regression analysis to identify factors associated with COVID-19 incidence and recovered cases are presented in Table 2. The predictors significantly associated with the COVID-19 incidence in both the linear and negative binomial regressions were population density, household crowding, obesity prevalence in women, and adult literacy; access to basic handwashing facilities at home was significantly associated with the COVID-19 incidence per 100,000 people only in the linear regression model. There was a positive association of COVID-19 incidence rate with the population density (RR = 1.38; 95% CI: 1.09, 1.76), household crowding (RR = 1.04; 95% CI: 1.01, 1.06), and the prevalence of obesity in women (RR = 1.07; 95% CI: 1.02, 1.13), where higher values of these variables were associated with a higher number of detected cases. In contrast, those with higher levels of literacy had a significantly lower incidence rate (RR = 0.97; 95% CI: 0.96, 0.99). When the analysis was continued with the outcome variable of recovered cases, population density (RR = 1.63; 95% CI: 1.18, 2.24) and adult literacy (RR = 1.03; 95% CI: 1.00, 1.06) were positively associated with an increased number of recovered cases (Table 2). When COVID-19 mortality was assessed, variables significantly associated with an increased COVID-19 mortality rate were population density, obesity in women, and smoking in men (Table 3). The linear regression model showed that geographic accessibility to healthcare facilities was negatively associated with deaths per 100,000 people.

Older age (aged ≥ 60 years) was not a significant factor in the multivariable models that included population density, sex ratio, obesity in women, smoking in men, access to a health facility, and per capita income as explanatory variables (Table 3). However, after omitting the lifestyle factors (smoking and obesity), the association between older age and COVID-19 mortality became significant in the negative binomial regression model (RR = 1.16; 95% CI: 1.01, 1.32). Further analysis revealed significant interactions between older age and sex ratio in COVID-19 mortality (*p* for interaction = 0.047).

Furthermore, we assessed the demographic and health-related indicators that had significant predictive power in our model. The correlation analysis (Appendix A) suggested that population density was the most important explanatory variable in the model. Indeed, it independently explained 76% of the variation in COVID-19 incidence (per 100,000 population) in the linear regression model (Appendix A). However, a significantly superior fit was obtained by adding the remaining explanatory variables (Table 2). This model accounted for 87% of the variation in the incidence. Population density continued to be the main determinant of the recovered cases and deaths per 100,000 people (Appendix A). The multiplicative interaction between population density and household crowding was not significant for deaths per 100,000 population (*p* = 0.960), and the interaction terms showed a marginal level of significance for cases and recovered cases per 100,000 population (*p* = 0.080 and 0.078, respectively).

We found a linear association between population density and COVID-19 cases, recovered cases, and deaths per 100,000 population (Figure 4). Figure 4A displays the relative increase in the incidence associated with the population density. In the study period, the incidence was higher in the areas with high density, and a rapid increase in the incidence was observed in the areas with a population density over 153.3 (log population density = 5.03). Figure 4C presents the number of deaths per 100,000 people and its association with population density. COVID-19 mortality risk was positively associated with population density; a significant increase in mortality was observed in the areas with a population density over 153.3.

Evaluation of the prediction model on the training and validation data using the linear regression model is shown in Figure 5. The scatter plots show that the model performed well in both datasets. However, the performance was better for training data, with adjusted R^2^ of 0.914, 0.894, and 0.584 for COVID-19 cases, recovered cases, and deaths (per 100,000 people), respectively. The scatter plots of the observed counts against the model-predicted counts using a negative binomial regression are shown in the Appendix A. The values of the adjusted Pseudo R^2^ indicated that the model performance was better for training data, which is consistent with the linear regression model.

## 4. Discussion

This is the first study to examine the risk factors associated with COVID-19-related health outcomes in Nepal. We found the districts that were vulnerable to the spread of COVID-19 in the study period due to high population density, the percentage of people living in crowded households, the percentage of people without access to basic handwashing facilities in their homes, and the percentage of adult literacy. Furthermore, higher COVID-19 transmission was associated with obesity in women. The mortality risk of COVID-19 was also generally higher in high-density areas. In addition, higher prevalences of obesity in women and smoking in men were associated with higher mortality rates, whereas access to a health facility was associated with a lower mortality rate. Population density was identified as the most important demographic variable associated with the large variation in COVID-19 transmission in our study.

Our finding of a positive association between population density and COVID-19 transmission is consistent with the existing literature [8,33]. The underlying mechanism for the association with population density is related to increased transmission of saliva, respiratory droplets, and or aerosol between individuals when people are in close physical proximity [34,35]. Furthermore, we found that household crowding was associated with higher COVID-19 transmission. Relatively few studies have examined the impact of household crowding, as opposed to household size. In the United States, an analysis of data from 91 counties in New York, New Jersey, and Connecticut found that people with more crowded households were more likely to contract COVID-19 infections [8]. Household crowding can increase the risk of exposure to coughs, sneezes, and food sharing, which have been considered as dominant risk factors for COVID-19 transmission [36]. Overcrowding is a common residential situation in Nepal due to small housing units, particularly for urban residents and large families. Therefore, improvement of overcrowded living conditions would reduce the high transmission of COVID-19.

We found a positive association between the lack of access to handwashing facilities and COVID-19 transmission. Limited access to handwashing facilities increases the risk of transmission from hands to eyes or mouth and may promote the spread and magnitude of the COVID-19 pandemic. Previous research showed that handwashing could reduce the transmission of respiratory viruses by 45−55% [37]. According to the WHO and UNICEF [38], the frequent washing of hands using water and soap could help contain the spread of COVID-19. Therefore, it is crucial to distribute hand sanitizers to those districts without access to handwashing facilities. Furthermore, our finding showed that areas with higher adult literacy rates were less vulnerable to COVID-19 spread. Potential reasons for the association might include higher rates of compliance with COVID-19 preventive measures among literate people, which may slow COVID-19 transmission [39].

Health-related lifestyle factors, such as obesity in women and smoking in men were significantly associated with the mortality rate in our study. Obesity has been identified as one of the key risk factors associated with COVID-19 deaths [40]. Our finding is consistent with a previous study that suggested a stronger risk of COVID-19 mortality in obese women than men [41]. Furthermore, we found a higher frequency of deaths in districts with a higher smoking prevalence in men. This finding is consistent with research showing higher risks of COVID-19-related death among current smokers [42]. In our study, the prevalences of obesity in men and smoking in women were not significantly associated with COVID-19 mortality; however, we cannot fully explore gender-specific differences in obesity and smoking for COVID-19 outcomes with this data, and further research should investigate how gender differences in COVID-19 outcomes vary regionally.

In addition, our study showed that the physical accessibility of medical services, which indicates the capability of a population to obtain health care services [43,44], was significantly associated COVID-19 mortality: areas with a lower percentage of health facilities within 30 min walking distance were likely to have a higher mortality rate. Previous research reported that limited or poor access to healthcare was associated with increased COVID-19 deaths [45]. People in areas with poor access to health facilities may delay receiving COVID-19 testing and diagnosis or even forgo being tested, and may, consequently, turn to medical care only in the advanced stages, which may result in poor outcomes.

Previous studies suggested that mortality due to COVID-19 was significantly higher in older people [46,47]. In our study, older age was significantly associated with a higher risk of COVID-19 mortality in the model adjusted for population density, sex ratio, access to a health facility, and per capita income. When the model was additionally adjusted for lifestyle factors, the association remained insignificant. In addition, the interactions between older age and sex ratio were significant in COVID-19 mortality. Our findings suggested a complex interplay of age, sex, and lifestyle factors in explaining the high mortality rate of COVID-19. Previous research highlighted a differential risk of COVID-19 mortality according to age, sex, and lifestyle factors [48,49]. However, further study using individual-level data are needed to confirm how age, sex, and lifestyle factors and their interactions contribute to the variations in the COVID-19 outcomes.

There are several limitations to this study. First, our study used district-level determinants. Therefore, the results of this study can only suggest associations between risk factors and COVID-19 outcomes at the district level but cannot be interpreted as the associations at the individual level. Second, the patient-level information was unavailable. Thus, the spread pattern of COVID-19 among specific sub-populations, such as among age, sex, and ethnicity subgroups, which might be associated differently with the other risk factors, could not be determined. Third, because of data limitations, we could not estimate district-specific data on the prevalence of clinical risk factors of COVID-19, including asthma, congestive heart failure, and cerebrovascular diseases. However, using the NDHS 2016 data, we provided available clinical risk correlates related to hypertension and obesity across districts. It should be noted that the estimates presented for these risk factors only apply to adults aged 15–49 years in the district, and not the entire district population. Finally, there is a possibility of bias due to the time elapsed between the current crisis and the collection of our data. However, the NPHC 2011 and NDHS 2016 are the main data sources in this study and are the most reliable data on the district-level demographic characteristics of Nepal, and major changes in the relative distributions of these measures are unlikely.

## 5. Conclusions

This study using district-level data from Nepal suggests that populations living in high-density areas may be more vulnerable to COVID-19 spread, as well as mortality. In addition, health- and sanitation-related population features, such as smoking prevalence, obesity rate, and access to a health facility and handwashing facility, may be contributing factors to the disparities in COVID-19 outcomes across districts. This study can provide a baseline for evaluating local level epidemic factors and designing policies for the control of local COVID-19 outbreaks.

## Figures and Tables

**Figure 1 ijerph-19-02659-f001:**
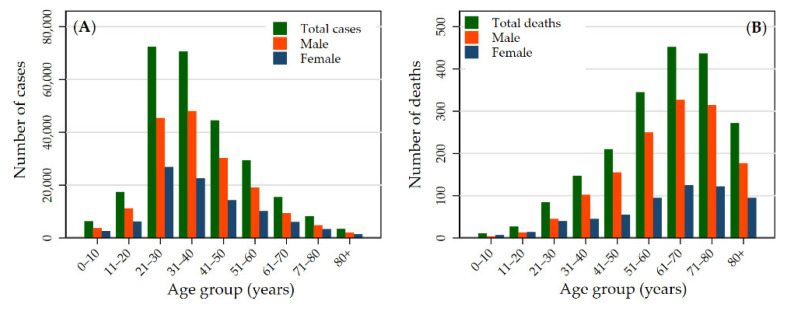
Age and gender distributions of confirmed COVID-19 cases (**A**) and deaths (**B**) in Nepal up to 22 January 2021.

**Figure 2 ijerph-19-02659-f002:**
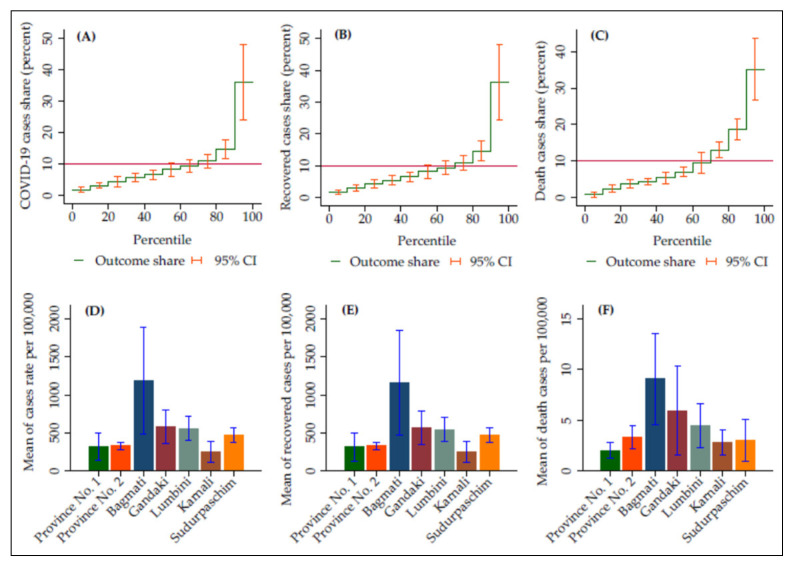
District incidence (**A**), recovery (**B**), and mortality (**C**) per 100,000 population, ranked by district percentile. Figures (**D**), (**E**), and (**F**) present the provincial distributions of COVID-19 outcomes.

**Figure 3 ijerph-19-02659-f003:**
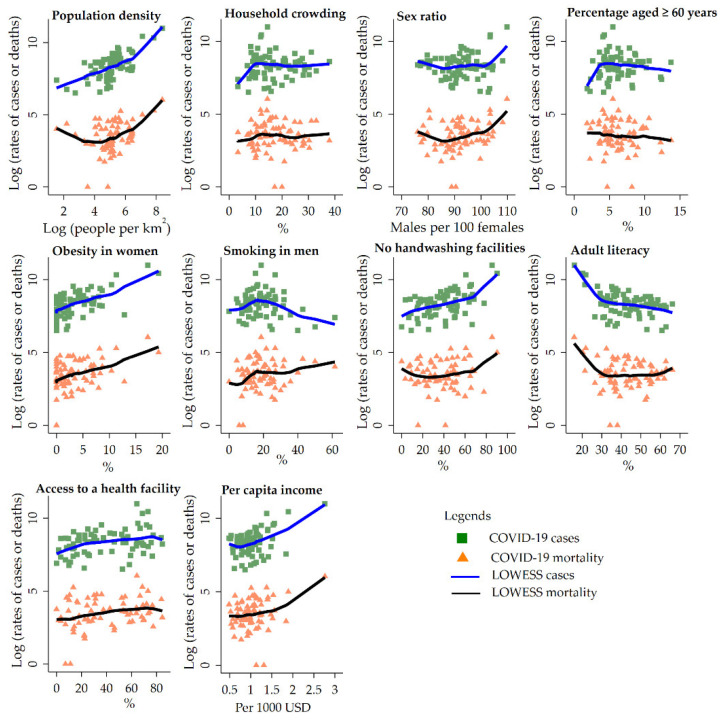
Bivariate analysis for the risk factors of COVID-19 cases and deaths per 100,000 population using LOWESS (locally weighted scatterplot smoothing) in the 73 districts.

**Figure 4 ijerph-19-02659-f004:**
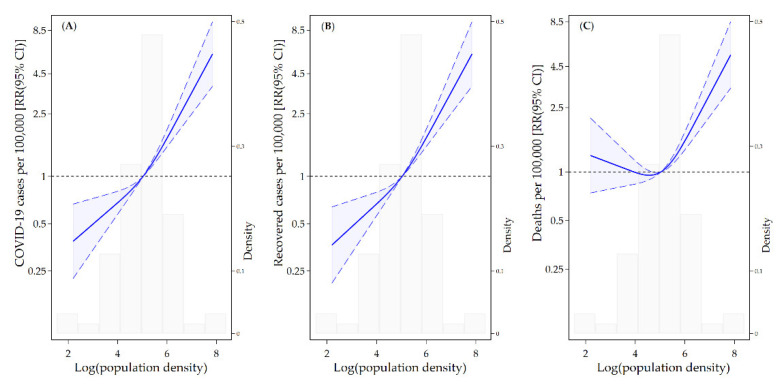
Associations of log-transformed population density with COVID-19 cases (**A**), recovered cases (**B**), and deaths (**C**) per 100,000 people using a restricted cubic spline with three knots. The solid line represents the relative risk (RR), and the long-dashed lines represent the confidence intervals. The reference population density for these plots (with RR fixed as 1.0) is 5.03. The histograms show the distribution of the log-transformed population density.

**Figure 5 ijerph-19-02659-f005:**
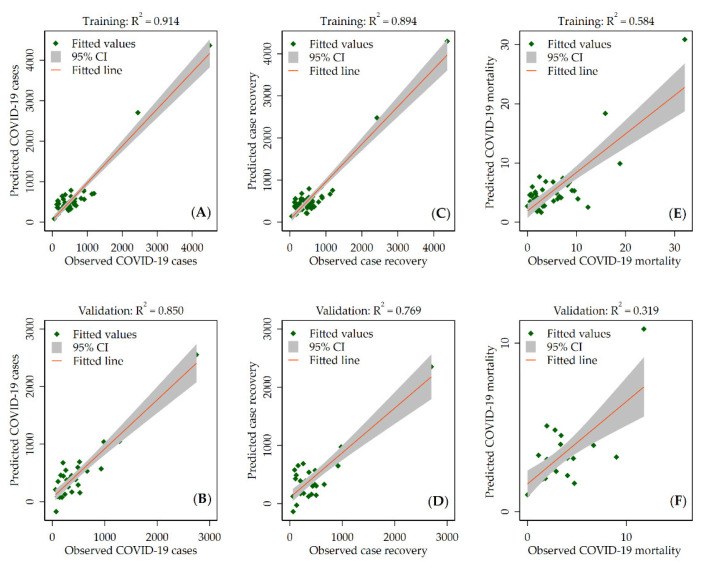
Scatter plot of the observed and predicted COVID-19 cases (**A**,**B**), recovered cases (**C**,**D**), and mortality (**E**,**F**) per 100,000 people for the training and validation datasets using the linear regression model. R^2^ is the adjusted R^2^ values.

**Table 1 ijerph-19-02659-t001:** District-level characteristics of the study population.

Characteristics	Mean	SD	Min	Max
COVID-19 infection characteristics				
Number of cases	3756.37	12,240.46	37	1035.23
Number of recovered cases	3677.27	11,902.28	35	1005.84
Number of deaths	27.18	86.66	0	738
Cases per 100,000 population	552.29	649.21	56.83	4499.26
Recovered cases per 100,000 population	540.26	635.53	53.76	4371.53
Deaths per 100,000 population	4.55	4.94	0.00	32.07
Case fatality rate (%) ^a^	1.20	1.46	0.00	9.09
Recovery rate (%) ^a^	97.21	2.53	84.09	99.72
Socio-demographic and health-related characteristics				
Population density (people per km^2^)	320.59	595.48	4.65	4415.80
Household crowding (%)	17.05	7.48	3.33	37.93
Sex ratio (number of males/females × 100)	91.99	7.77	76.02	109.84
Percentage aged ≥ 60 years	6.13	2.38	1.82	13.74
Obesity prevalence (%)				
Women	3.59	3.98	0.00	19.30
Men	1.86	2.12	0.00	8.43
Smoking prevalence (%)				
Women	5.36	5.37	0.00	20.93
Men	18.85	10.53	0.00	61.54
No access to handwashing facilities (%)	60.90	20.19	9.76	100
Adult literacy rate (%)	42.76	11.42	15.96	66.11
Access to a health facility (%)	39.85	24.70	0.00	84.62
Per capita income (USD)	1039.71	360.30	487.00	2764.00

^a^ Calculated by dividing the number of events by the total number of reported cases. USD, United States Dollar.

**Table 2 ijerph-19-02659-t002:** Multivariable linear regression and negative binomial regression analyses on COVID-19 case diagnosis and successful resolution of disease.

Variables	Linear Regression	Negative Binomial Regression
	β (95% CI)	RR (95% CI)
Incidence ^a^		
Population density	0.689 (0.571, 0.806) ***	1.38 (1.09, 1.76) ^c^ **
Household crowding (%)	15.18 (5.86, 24.51) **	1.04 (1.01, 1.06) **
Obesity prevalence in women (%)	45.74 (13.87, 77.60) **	1.07 (1.02, 1.13) **
Smoking in men (%)	3.00 (−1.65, 7.66)	1.00 (0.98, 1.01)
No access to handwashing facilities (%)	5.52 (0.698, 10.35) *	1.00 (0.99, 1.01)
Adult literacy (%)	−15.32 (−22.43, −8.22) ***	0.97 (0.96, 0.99) **
Percentage aged ≥ 60 years	−11.86 (−34.61, 10.90)	0.98 (0.93, 1.03)
R^2^/McFadden’s Pseudo R^2^	0.874	0.047
Recovered cases ^b^		
Population density	0.752 (0.618, 0.887) ***	1.63 (1.18, 2.24) **
Sex ratio	3.92 (−13.59, 21.44)	1.01 (0.98, 1.04)
Adult literacy (%)	16.42 (0.744, 32.09) *	1.03 (1.00, 1.06) *
Access to a health facility (%)	−0.073 (−2.83, 2.68)	1.00 (0.99, 1.01)
Per capita income (USD)	0.198 (−0.104, 0.500)	1.08 (0.53, 2.22) ^d^
Percentage aged ≥ 60 years	−7.42 (−39.80, 24.97)	0.99 (0.92, 1.07)
R^2^/McFadden’s Pseudo R^2^	0.826	0.037

* *p* < 0.05, ** *p* < 0.01, *** *p* < 0.001. ^a^ Dependent variable: cases per 100,000 population in the linear regression model and the number of confirmed cases in the negative binomial regression model. ^b^ Dependent variable: recovered cases per 100,000 population in the linear regression model and the number of recovered cases in the negative binomial regression model. ^c^ For every 1000 population. ^d^ For every thousand dollars increase in per capita income.

**Table 3 ijerph-19-02659-t003:** Multivariable linear regression and negative binomial regression analyses on COVID-19 mortality.

Variables	Linear Regression ^a^	Negative Binomial Regression ^b^
	β (95% CI)	RR (95% CI)
Population density	0.006 (0.004, 0.007) ***	1.42 (1.07, 1.88) ^c^ *
Sex ratio	−0.047 (−0.191, 0.096)	0.99 (0.97, 1.03)
Obesity prevalence in women (%)	0.293 (−0.033, 0.618) ^+^	1.06 (1.00, 1.14) *
Smoking in men (%)	0.073 (−0.002, 0.148) ^+^	1.01 (1.00, 1.03) *
Percentage aged ≥ 60 years	−0.107 (−0.506, 0.293)	0.97 (0.88, 1.06)
Access to a health facility (%)	−0.040 (−0.080, −0.001) *	1.00 (0.99, 1.004)
Per capita income (USD)	0.001 (−0.003, 0.004)	1.20 (0.54, 2.65) ^d^
R^2^/McFadden’s Pseudo R^2^	0.566	0.030

^+^*p* < 0.1, * *p* < 0.05, *** *p* < 0.001. ^a^ Dependent variable: deaths per 100,000 population. ^b^ Dependent variable: number of deaths due to COVID-19. ^c^ For every 1000 population. ^d^ For every thousand dollars increase in per capita income.

## Data Availability

The data presented in this study are available upon request from the corresponding author.

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
