# Peer review of "District-Level Risk Factors for COVID-19 Incidence and Mortality in Nepal"

_ijerph, 2022, doi:10.3390/ijerph19052659_

Round 1
Reviewer 1 Report
Please see the provided pdf that lists specific comments, then provides detailed suggestions on refining the English and improving clarity on an annotated version of the manuscript.

Reviewer 2 Report
This study deals with Nepals SARS-CoV-2 (COVID-19) cases, recovery and mortality levels and assesses factors associates with each of them at the district level. The study is of high relevance as we still are learning about this novel disease and its effects at local, national and global levels. However, there are things the authors need to address to in order to strengthen their work.
Methods:
- It is not clear how the Gaussian and negative binomial models were set up. Have you added the population size as an offset terms? If not how are you controlling for differences in population sizes among districts?
- In order to see how good the fitted models are, it would be good to set aside part of the data for validating the models. This does not seem to have been the case. This would include comparison of the goodness of fits for each model used.
- For a better grasp of why proportion of seniors did not show up as a significant contributor, I would add a table or plot of the age distribution of deaths. That would indicate if age really did not matter or whether the linear regression for deaths had a collinearity effect which resulted in non-significant effect of proportion of seniors.
Results
- One useful output missing is a scatter of estimated (including 95%CI) by observed cases both for the Gaussian and Negative Binomial models. This way one would be able to see how these models have performed fitting to the data. Especially with analysis of models by separating training and validating datasets, it would be desirable to have a section showing how estimates and data compare for each model considered.
Other comments:
These data source were used for corresponding data on each variable: however there is no details on how these were extracted and used at district level.
NDHS: obesity, hypertension, handwashing facility, access to health facility, cigarette smoking,
UNDP: Literacy rate
Nepal Human Development Report: per capita income
These variables come at geographic resolution larger than district. The authors need to clarify how they converted (downscaled) values to district level.
Lines 279-282:
What is age factor? I presume it could be percentage >60 years, but would be advisable to use a consistent term, in this case the latter.
Round 2
Reviewer 1 Report
Comments on the revised manuscript:
The authors have greatly improved the manuscript – it is far easier to read and clearer.
I have made a few quite minor suggestions around phrasing in green font or comments in the attached pdf. I would ask the authors to check that my suggestions are in keeping with the intention of the sentence.
There were only two key points that needed clarification:
- The description of stratification is hard to understand (see highlighted text at line 115, and immediately after it).
Should the comma be a fullstop?
Are you saying that the country was first stratified by district then by wards? From the description as given, it seems like there are 4-5 levels of stratification rather than the 2-3 stated by the authors? i.e.
i) district
ii) urban and rural areas
iii) wards
iv) enumeration area (urban sites only)
v) households
I suspect this is purely a matter of clarifying the English expression.
- One lesser point: Can you explain to the reader (within the manuscript) why and how 7 provinces were chosen out of the 73 provinces for further analysis. You could do this either in the methods, if it was planned in advance, or in the results (e.g. around line 293), if it arose as a product of the data analysis.

Reviewer 2 Report
Thank you for addressing most of my questions. I still have difficulty accepting this statement as is.
Although the highest number of cases was observed in the elderly population, age ≥ 60 was not a significant factor in our analysis.
Senior age seems to be an important factor for death as can be seen in Figure 1B, especially when considering the low number of cases for that age category overall.
What I am concerned about is a potential for misinterpretation of the effect of senior age on mortality. The data seem to suggest elderly people at higher risk of death. The authors need to explain their results in this context. The reasons why the model would not show significant effect of proportion of senior population should be discussed in more details.
